# Profiling the lymphoid-resident T cell pool reveals modulation by age and microbiota

Aurélie Durand[1], Alexandra Audemard-Verger[1], Vincent Guichard[1,2], Raphaël Mattiuz[1], Arnaud Delpoux [1], Pauline Hamon[3], Nelly Bonilla[1], Matthieu Rivière[1], Jérôme Delon[1], Bruno Martin[1], Cédric Auffray[1], Alexandre Boissonnas[3] & Bruno Lucas[1]

Despite being implicated in non-lymphoid tissues, non-recirculating T cells may also exist in secondary lymphoid organs (SLO). However, a detailed characterization of this lymphoid-resident T cell pool has not yet been done. Here we show that a substantial proportion of CD4 regulatory (Treg) and memory (Tmem) cells establish long-term residence in the SLOs of specific pathogen-free mice. Of these SLOs, only T cell residence within Peyer's patches is affected by microbiota. Resident CD4 Treg and CD4 Tmem cells from lymph nodes and non-lymphoid tissues share many phenotypic and functional characteristics. The percentage of resident T cells in SLOs increases considerably with age, with S1PR1 downregulation possibly contributing to this altered homeostasis. Our results thus show that T cell residence is not only a hallmark of non-lymphoid tissues, but can be extended to secondary lymphoid organs.

[1] Paris Descartes Université, Sorbonne Paris Cité, Institut Cochin CNRS UMR8104, INSERM U1016, 27 rue du Faubourg Saint-Jacques, 75014 Paris, France.
[2] Paris Diderot Université Sorbonne Paris Cité, 75013 Paris, France. [3] Pierre et Marie Curie Université (UPMC), Sorbonne Universités, INSERM U1135, CNRS ERL8255, Centre d'Immunologie et des Maladies Infectieuses, 91 Boulevard de l'Hôpital, 75013 Paris, France. Aurélie Durand and Alexandra Audemard-Verger contributed equally to this work.  Correspondence and requests for materials should be addressed to B.L. (email: bruno.lucas@inserm.fr)

It has been hypothesized that peripheral T cells recirculate continuously between lymphoid organs to scan antigen presenting cells (APC) for the presence of foreign antigens. Such a model has been challenged in the last decade by numerous reports demonstrating the existence of T cells residing in non-lymphoid tissues, mostly memory CD8 T cells (CD8 Tmem cells)[1]. Indeed, results from tissue graft and parabiosis experiments have demonstrated the resident nature of a substantial proportion of CD8 Tmem cells found in several tissues, including skin, intestine, brain, lungs and salivary glands[1]. A study showed that, for a given specificity, memory T cells residing in non-lymphoid tissues outnumber their circulating cell-counterparts[2]. Although the existence of tissue-resident memory T cells is best documented for CD8 T cells, non-recirculating subsets of CD4 memory T (CD4 Tmem) cells have also been described[3]. Finally, tissue-resident regulatory CD4 T (CD4 Treg) cells have been found in multiple tissues, including the skin, muscle, lungs, adipose tissue, and intestine[4,5]. Resident memory T cells may represent a first line defense against pathogens at sites of infection, whereas resident CD4 Treg cells may ensure tissue integrity by dampening T cell responses to self antigens and commensal bacteria antigens, and by controlling crosstalks between immune and non-immune cells[6–8]; for example, skin resident CD4 Treg cells crosstalk with hair follicle stem to modulate skin wound healing and hair regeneration[9,10].

Resident T cells have been extensively studied within non-lymphoid tissues. However, there is now evidence that resident T cells might also exist within secondary lymphoid organs (SLO)[8]. In humans, it was shown that, in spleen, lymph nodes (LN), and tonsils, a significant fraction of CD4 and CD8 Tmem cells phenotypically resembles resident T cells within non-lymphoid tissues, and that, at least for CD8 T cells, they included cells with defined specificity for EBV and CMV[11–13]. The existence of a subset of effector CD4 Tmem cells retained in mouse SLOs that accumulated after immunization or in response to chronic antigen exposure has been suggested using photoconvertable fluorescence reporters[14–17], with the implicated resident T cell subsets including follicular helper CD4 T cells[15,17] and populations of innate-like αβ and γδT cells expressing CCR6 and high surface levels of CD127[16]. Retention of CD8 Tmem cells within draining mediastinal LNs after lung infections and within spleen and LNs after LCMV acute infection in mice has also been shown[18,19].

We and others have recently shown that interactions between TCR and self peptides/self MHC class II complexes help retain, at least temporarily, CD4 T cells in mouse LNs[20–22]. Using two different experimental approaches, here we show the long-term residence of a substantial proportion of CD4 Treg and CD4 Tmem cells in the SLOs of specific pathogen-free (SPF) mice. By contrast, CD8 Tmem cells are retained only in Peyer's patches. Microbiota has important function in T cell residence in Peyer's patches, but only a minor one, if any, in LNs. LN-resident CD4 Treg and CD4 Tmem cells share many phenotypic and functional characteristics, including a core transcriptional profile, with their counterparts from non-lymphoid tissues. In particular, S1PR1 downregulation may represent the main mechanism accounting for T cell residency within SLOs. Strikingly, T cell residence increases with age, with the majority of CD4 Treg and Tmem cells in the LNs being resident but not circulating T cells in old mice.

## Results

**A proportion of T cells is retained in the SLOs of SPF mice.** To study T cell residence within SLOs, we first generated CD45.1/CD45.2 parabiotic mice and analyzed them 4 weeks after surgery (Fig. 1a). Throughout this study, CD4 Treg cells were defined as Foxp3$^+$CD4$^+$CD8α$^-$TCRβ$^+$ cells, CD4 Tmem cells as CD44$^{hi}$Foxp3$^-$CD4$^+$ CD8α$^-$TCRβ$^+$ cells, and naive CD4 T cells as CD44$^{-/low}$Foxp3$^-$CD4$^+$CD8α$^-$TCRβ$^+$ cells (Fig. 1b). CD44 expression was also used to discriminate between naive and memory CD8 T cells (Fig. 1b). Interestingly, unlike naive CD4 and CD8 T cells, CD45.1$^+$ and CD45.2$^+$ CD4 Tmem and CD4 Treg cells were not equally distributed in all studied SLOs (including Peyer's patches and the spleen) with a consistent enrichment in host-derived cells (CD45.1$^+$ cells in the CD45.1 parabiont and CD45.2$^+$ cells in the CD45.2 parabiont, Fig. 1c–e). Such a phenomenon was observed for CD8 Tmem cells only in Peyer's patches. Similar results were found when parabiotic mice were studied 2 months after surgery (Supplementary Fig. 1a). We have recently shown that Ly-6C expression allows to discriminate between memory and naive CD4 Treg cells with Ly-6C$^-$ cells corresponding to memory CD4 Treg cells[22]. Using this marker, we then tested whether the unequal redistribution of CD4 Treg cells between the two parabionts was differential between the naive and memory subpopulations. Interestingly, we found that among CD4 Treg cells, only Ly-6C$^-$ CD4 Treg cells were enriched in cells deriving from their host in all SLOs tested (Fig. 1f). Similar results were obtained by using, as proposed by Smigiel et al.[23], the expression level of CD44 to discriminate between naive (CD44$^{low}$) and memory (CD44$^{high}$) CD4 Treg cells (Supplementary Fig. 1b). Altogether, these results suggest that a significant proportion of memory regulatory and conventional T cells are residing for at least several weeks within the SLOs of SPF mice.

To confirm our hypothesis, we then blocked T cell entry into LNs and Peyer's patches through the i.p. injection of mAbs specific to VLA-4 $and LFA-1 as previously described[20], and first studied the cells remaining in these SLOs 24 h after initiating the treatment (Fig. 2a). At this time-point, most CD4 and CD8 T cells have exited the LNs and accumulated in the spleen (Supplementary Fig. 2a, b). Interestingly, CD4 Treg and CD4 Tmem cells were greatly enriched in the remaining LN CD4 T cells (Fig. 2b). Consistent with the results obtained in parabiotic mice, 24 h after mAb injection, enrichment in CD8 Tmem cells was clearly observed within Peyer's patches (Fig. 2c). We then blocked T cell entry into LNs for 7 days by repeating the mAb treatment every 2 days (Fig. 2d). This protocol led to a gradual increase of the proportion of CD4 Treg and CD4 Tmem cells within the CD4 T cells remaining in LNs, and a concomitant decrease of the percentage of naive CD4 T cells (Fig. 2e). More precisely, whereas naïve T cells pursued exiting LNs at least until day 4, the absolute numbers of LN CD4 Treg and Tmem cells remained almost stable (Fig. 2f and Supplementay Fig. 2c).

Thus, using two different experimental approaches, we obtained data suggesting strongly that a substantial proportion of CD4 Treg cells and of CD4 Tmem cells are not continuously recirculating through the SLOs of SPF mice, but rather reside for long periods of time in them. This unexpected result seems to apply to CD8 Tmem cells, in Peyer's patches only.

**The proportion of LN-resident CD4 T cells increases with age.** Surprisingly, whereas the proportions of CD4 Treg cells and Tmem cells strongly increase with age in SLOs, the composition of the blood remains quite unchanged (Fig. 3a). This led us to suggest that the qualitative changes with age of the CD4 T cell pool in SLOs may reflect an increase with age in the proportion of resident CD4 Treg cells and CD4 Tmem cells. To test this hypothesis, we applied the mAb treatment described above to young and old mice and analyzed the results as a function of their age (Fig. 3b). Two days after treatment, whereas the CD4 Treg and Tmem cells remaining in LNs represent <25% of the corresponding pools in 4-week-old animals, these proportions rise to

50–75% in 1-year-old mice. For each experiment, we then cal-culated the proportion of circulating CD4 Treg cells and of CD4 Tmem cells among LN CD4 T cells by subtracting the numbers of cells recovered 2 days after treatment from the total number of

these cells recovered from LNs (Fig. 3c, d; see Experimental procedures). Interestingly, by doing so, we observed that the corrected proportion of circulating CD4 Treg cells did not vary with age in LNs as observed in the blood. The increase in the

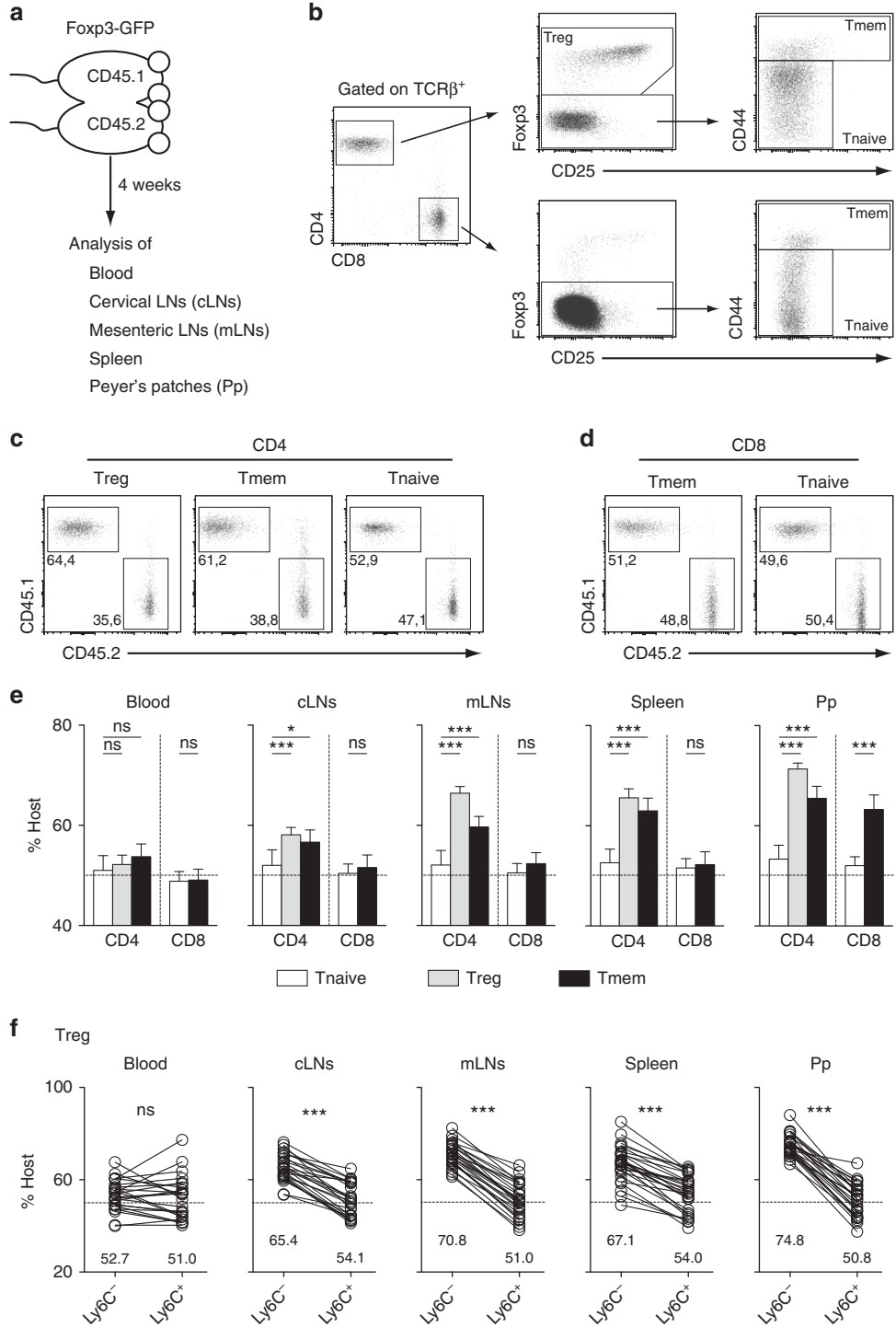

**Fig. 1** Analysis of mouse parabionts reveals the long-term residence of T cell populations in SLOs. Four weeks after parabiosis surgery, blood, cervical LNs (cLNs), mesenteric LNs (mLNs), spleen, and Peyer's patches (Pp) were recovered and analyzed. **a** Diagram illustrating the experimental model. **b** Gating strategy to define naïve (Tnaive), memory (Tmem) and regulatory (Treg) CD4 and CD8 T cells. **c** CD45.1/CD45.2 dot-plots are shown for mLN CD4 Treg, Tmem and Tnaive cells from a representative CD45.1 parabiont. **d** CD45.1/CD45.2 dot-plots are shown for mLN CD8 Tmem and Tnaive cells from a representative CD45.1 parabiont. **e** Proportions of host cells (CD45.1+ for the CD45.1 parabiont and CD45.2+ for the CD45.2 parabiont) among the indicated CD4 T cell and CD8 T cell subsets recovered from blood, cLNs, mLNs, Spleen, and Pp are shown for at least ten parabiotic pairs from three independent experiments (mean ± SEM; paired *t*-test). **f** Proportions of host cells among Ly-6C− and Ly-6C+ CD4 Treg cells. Each pair of dots represents an individual mouse (paired *t*-test). *$p < 0.05$, ***$p < 0.001$. ns not significant

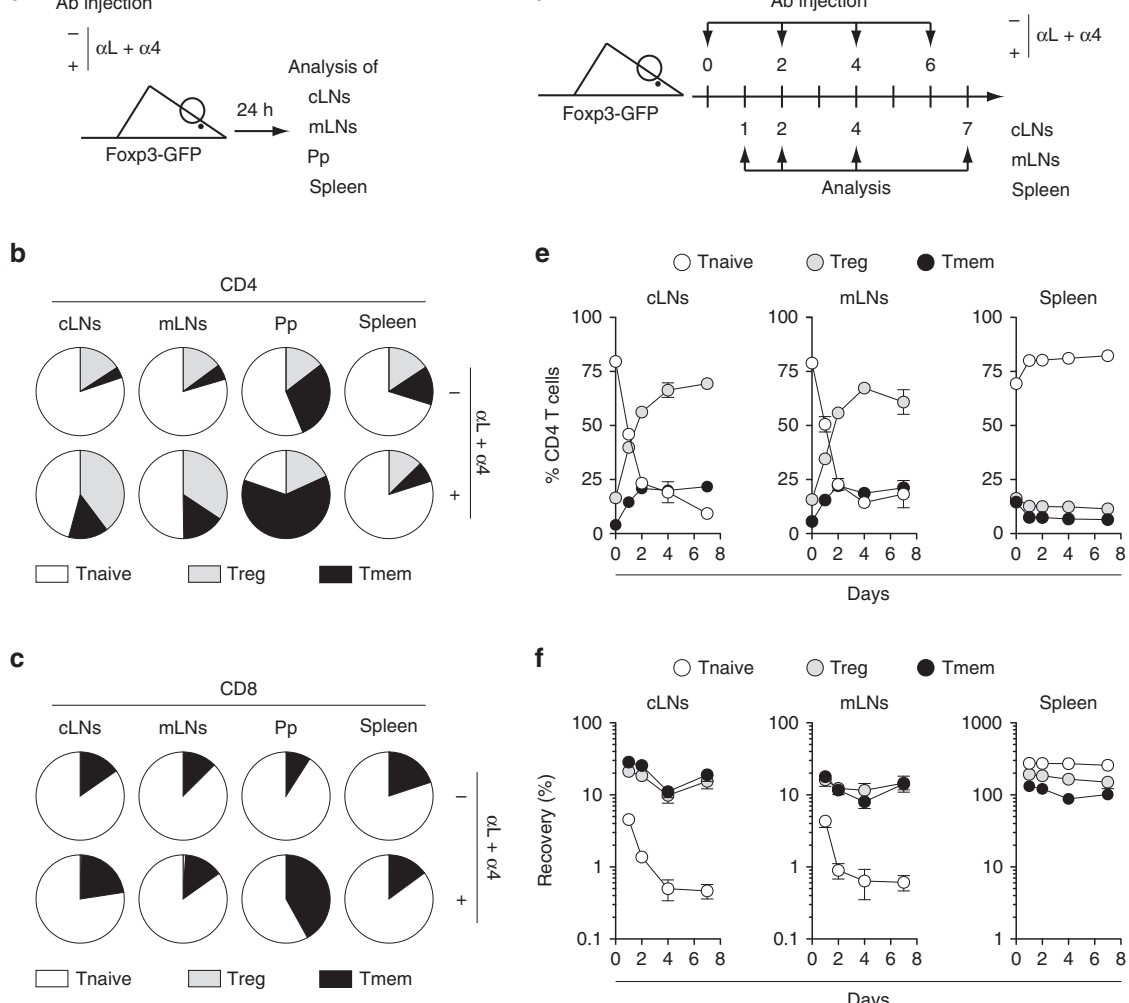

**Fig. 2** Analysis of residual T cells after blocking T cell entry into LNs and Peyer's Patches. **a–c** 6–12-week-old C57BL/6 Foxp3-GFP mice were injected or not i.p. with 200 μg of anti-LFA-1 (αL) and anti-VLA-4 (α4) Abs. Twenty-four hours later, cLNs, mLNs, Pp, and spleen were harvested and analyzed. **a** Diagram illustrating the experimental model. **b** Distribution of Treg, Tmem, and Tnaive cells among CD4 T cells in the indicated SLOs of treated or untreated mice. **c** Distribution of Treg, Tmem, and Tnaive cells among CD8 T cells in the indicated SLOs of treated or untreated mice. **d**, **e** 6–12-week-old C57BL/6 Foxp3-GFP mice were injected or not i.p. with 200 μg of anti-LFA-1 (αL) and anti-VLA-4 (α4) Abs every 2 days from day 0 to day 6 and SLOs were recovered for analysis at various time-points. **d** Experimental model. **e** Percentages of Treg, Tmem, and Tnaive cells among CD4 T cells are shown as means ± SEM for the indicated SLOs. **f** Relative absolute numbers of CD4 Treg, Tmem, and Tnaive cells in cLNs, mLNs, and spleen are shown. The percentage of recovery was calculated by dividing the absolute numbers in treated mice by the mean absolute number obtained in untreated animals. Data are means ± SEM for at least three independent experiments. Mouse clip arts were generated in ref. [21]

proportion of LN CD4 Tmem cells was also reduced when the correction factor was applied. Thus, the proportion of LN-resident CD4 Treg and Tmem cells increases with age to represent the majority of these cells in old mice.

**LN-resident CD4 T cells exhibit an effector phenotype**. We then analyzed the phenotype of CD4 T cells remaining within LNs 2 days after anti-VLA-4 and anti-LFA-1 mAb injection in 6–12-week-old mice. For simplicity's sake, we will refer to these cells as "day 2 CD4 T cells". Day 2 CD4 Treg and CD4 Tmem cells exhibited decreased expression of CD62L, CD127, and CCR7 when compared to their counterparts from untreated mice in both peripheral LNs (pLN; Fig. 4a) and mesenteric LNs (mLN; Supplementary Fig. 3a). In line with this effector phenotype, day 2 CD4 Treg cells from both pLNs and mLNs contained a higher proportion of cycling cells (as assessed by Ki-67 staining) than CD4 Treg cells from untreated mice (Fig. 4b and Supplementary Fig. 3b). A weak increase in the percentage of Ki-67-expressing

cells was observed for CD4 Tmem cells in mLNs only. Upon a 2 h re-stimulation with PMA and ionomycin, about three times more day 2 LN CD4 Treg cells were able to produce IL-10 than LN CD4 Treg cells from untreated animals (Fig. 4c and Supplementary Fig. 3c). Similarly, day 2 LN CD4 Tmem cells were enriched in cytokine-producing cells (especially IL-17) when compared to control LN CD4 Tmem cells. We then divided the absolute number of day 2 CD4 T cells producing a given cytokine by the number of CD4 T cells from untreated mice producing the same cytokine. Interestingly, the results obtained suggest that nearly all IL-10+ CD4 Treg cells and almost all IL-17+ CD4 Tmem cells are in fact LN-resident T cells (Fig. 4d and Supplementary Fig. 3d). Of note, integrin engagement by monoclonal antibodies prior to re-stimulation with PMA and ionomycin does not potentiate the ex vivo ability of T cells to produce cytokines (Supplementary Fig. 4).

Finally, we studied more extensively the phenotype of day 2 CD4 Treg cells (Supplementary Fig. 5). We first confirmed the

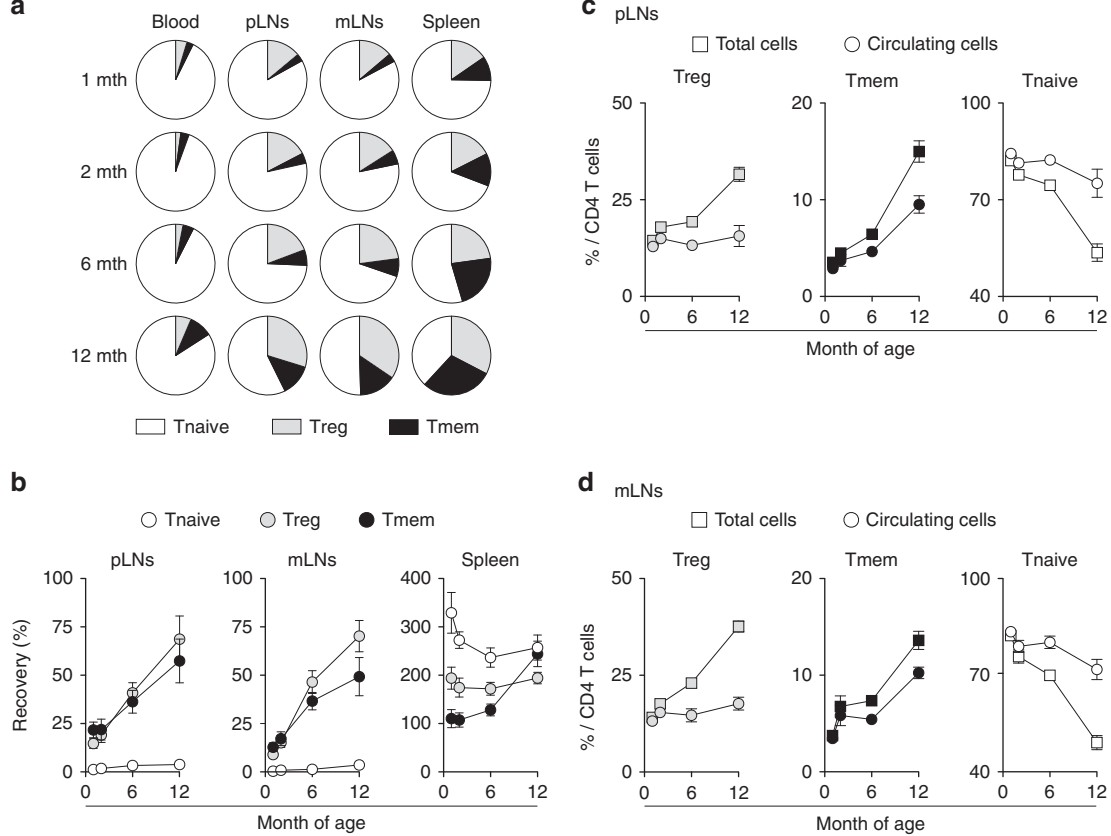

**Fig. 3** T cell residence in SLOs increases with age in SPF mice. **a** Distribution of Treg, Tmem, and Tnaive cells among CD4 T cells in blood, pLNs (pooled superficial cervical, axillary, brachial, and inguinal LNs), mLNs, and spleen of 1–12-month-old C57BL/6 Foxp3-GFP mice. **b** C57BL/6 Foxp3-GFP mice of various ages were injected or not i.p. with 200 µg of anti-LFA-1 (αL) and anti-VLA-4 (α4) Abs. Forty-eight hours later, blood, pLNs, mLNs, and spleen were harvested and analyzed. Relative absolute numbers (recovery) of CD4 Treg, Tmem, and Tnaive cells in the indicated SLOs are shown as means ± SEM as a function of mouse age. **c**, **d** Percentages of Treg, Tmem, and Tnaive cells among total or "circulating" CD4 T cells are shown as means ± SEM for the indicated SLOs (see the experimental procedures for calculations)

memory phenotype of these cells as they were expressing high surface amounts of CD44 and were not expressing Ly-6C[22,23]. We then examined the expression of molecules known to play a role in the suppressive capacities of CD4 Treg cells. All tested molecules, except CD25, were found to be upregulated in day 2 LN CD4 Treg cells when compared to LN CD4 Treg cells from control mice (Supplementary Fig. 5).

Altogether, our results show that day 2 LN CD4 Treg and Tmem cells have an effector phenotype that correlates with their increased ability to produce cytokines upon re-stimulation.

**Microbiota affects T cell residence in Peyer's patches.** We hypothesized that CD4 T cells residing in SLOs would be receiving more TCR signals than the bulk of CD4 T cells and that these signals might convert/maintain them in an effector state. The lack of Ly-6C expression of day 2 CD4 Treg cells fits with this hypothesis (Supplementary Fig. 5). Indeed, we have recently shown that Ly-6C downregulation by peripheral memory/effector CD4 Treg cells resulted from interactions with self MHC class II molecules[22]. To go further, we studied the expression of CD5 and Nur77, two molecules whose levels of expression have been shown to reflect TCR engagement[24–26]. Both molecules were found to be upregulated in day 2 CD4 Treg and CD4 Tmem cells in both pLNs and mLNs (Fig. 5a, b and Supplementary Fig. 6a, b). Collectively, these results strongly suggest that day 2 CD4 Treg and Tmem cells interact more strongly with MHC class II-expressing cells in SLOs than their circulating counterparts.

To determine the relative contribution of microbiota- and self antigen-derived peptides in the retention of T cells within SLOs, we first studied germ free mice (Fig. 5c). The total cell numbers in pLNs and mLNs were comparable between Germ free and SPF mice whereas the cellularity of Peyer's patches was strongly reduced in germ free mice when compared to SPF mice (Fig. 5d). In agreement, the recovery of day 2 CD4 Treg and Tmem cells was only slightly reduced in pLNs and not affected in mLNs (Fig. 5e). By contrast, 2 days after mAb injection, cell recovery was strongly diminished in the Peyer's patches of germ free mice when compared to control mice for both CD4 and CD8 Tmem cells (Fig. 5f). Although not significant, a similar tendency seems to apply to CD4 Treg cells. These results were largely confirmed in mice treated with a combination of antibiotics for 4 weeks before blocking T cell entry into LNs and Peyer's patches (Supplementary Fig. 6c–f). In this setting, although the total cellularity as well as the absolute numbers of CD4 Treg and Tmem cells were slightly reduced in the pLNs of antibiotic treated mice, the absolute numbers of day 2 CD4 Treg and Tmem cells were not affected by the antibiotic treatment in both pLNs and mLNs. Two days after mAb injection, cell recovery was strongly diminished in Peyer's patches for CD4 Treg cells and for both CD4 and CD8 Tmem cells (Supplementary Fig. 6f). Microbiota thus seems to have an important impact on T cell residence in Peyer's patches but only a small or no role in LNs.

**Transcriptional signature of LN-resident CD4 T cells.** We then decided to compare the transcriptome of day 2 LN CD4 Treg and

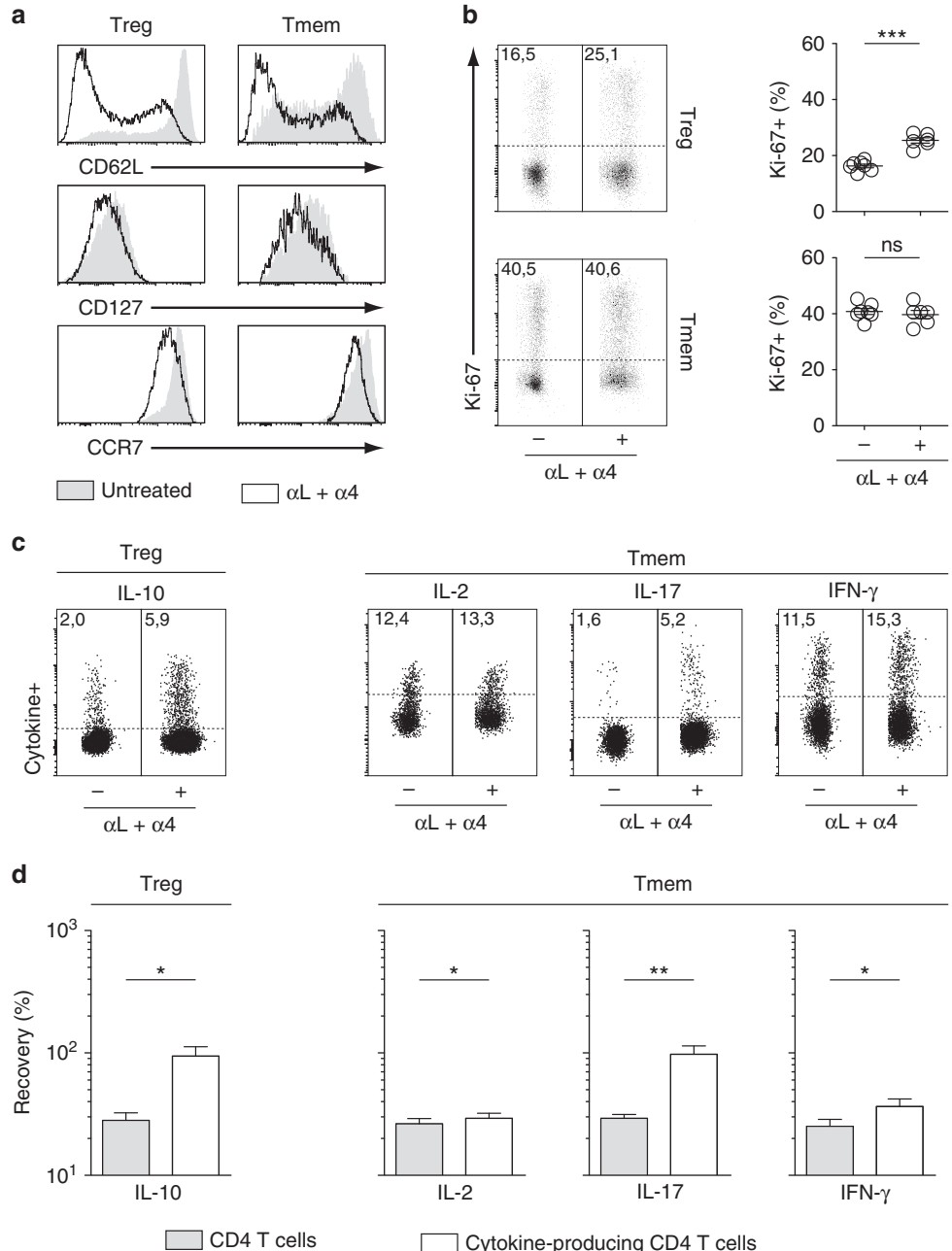

**Fig. 4** pLN-resident CD4 T cells exhibit an effector phenotype. 6–12-week-old C57BL/6 Foxp3-GFP mice were injected or not i.p. with 200 μg of anti-LFA-1 (αL) and anti-VLA-4 (α4) Abs. Forty-eight hours later, pLNs were harvested and analyzed. **a** CD62L, CD127, and CCR7 fluorescence histograms of CD4 Treg and CD4 Tmem cells from a representative treated and a representative control C57BL/6 Foxp3-GFP mouse. **b** Representative Ki-67 expression by CD4 Treg and CD4 Tmem cells from treated and control mice. Quantification is shown on the right side of the panel (unpaired *t*-test). Each dot represents an individual mouse. **c** Representative IL-10 expression by CD4 Treg cells and representative IL-2, IL-17, and IFN-γ expression by CD4 Tmem cells from treated and control mice. **d** Quantification of cytokine production by CD4 Treg and Tmem cells is shown as means ± SEM with paired *t*-test. The percentage of recovery for a given cytokine was calculated by dividing the absolute numbers of cells expressing this cytokine in treated mice by the mean absolute number of cells expressing this same cytokine in untreated animals (white bars). The gray bars correspond to the recovery of total CD4 Treg or CD4 Tmem cells in the same samples. *$p < 0.05$, **$p < 0.01$, ***$p < 0.001$. ns not significant

Tmem cells to the transcriptome of the corresponding CD4 T cells from unmanipulated animals. With a fold change cut-off of ≥1.5 and a *p*-value cut-off of ≤0.05, we found that 398 genes were differentially expressed in day 2 CD4 Treg cells and 381 in day 2 CD4 Tmem cells, when compared to their cell-counterparts from control mice (Supplementary Fig. 7a).

We then further compared our transcriptomic signatures to transcriptomic signatures from the literature characterizing CD4

Treg and Tmem cells residing in non-lymphoid tissues as done recently by Haskett et al.[27]. Data sets were filtered to common probes. Interestingly, day 2 LN CD4 Treg and Tmem cells shared the expression of many genes specific to resident CD4 Treg and Tmem cells from non-lymphoid tissues, such as muscle[28], (Fig. 6a, b) and visceral adipose tissue (VAT)[29], (Supplementary Fig. 7b, c). Indeed, most genes overexpressed in CD4 Treg cells from muscle and VAT were found to be upregulated in day 2 LN CD4

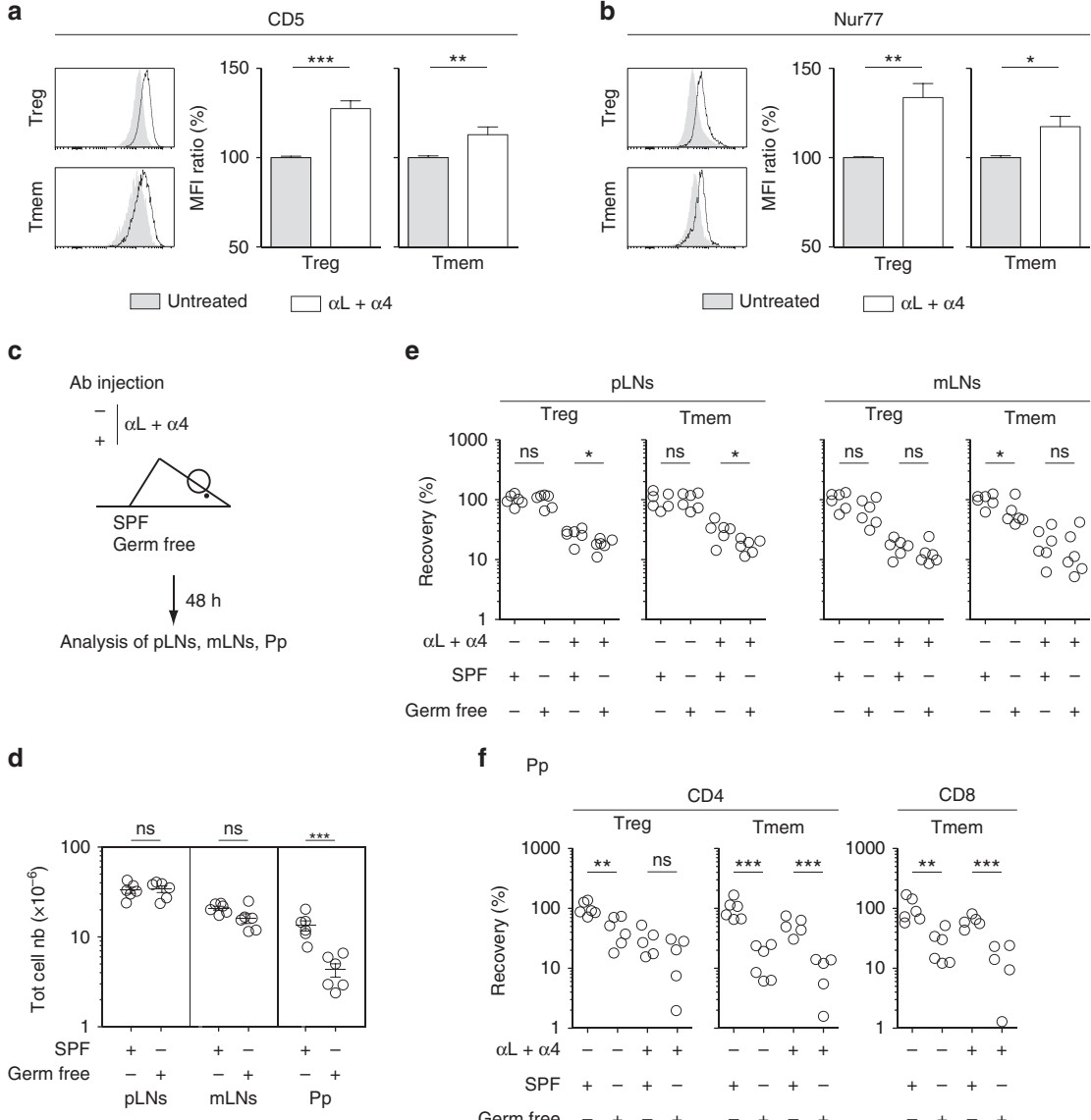

**Fig. 5** Assessing the relative role of self antigens and microbiota in T cell residence in SLOs. **a–c** 6–12-week-old C57BL/6 Foxp3-GFP mice were injected or not i.p. with 200 µg of anti-LFA-1 (αL) and anti-VLA-4 (α4) Abs. Forty-eight hours later, peripheral lymph nodes (pLN) were harvested and analyzed. CD5 (**a**) and Nur77 (**b**) fluorescence histograms of CD4 Treg and CD4 Tmem cells from the pLNs of a representative treated and a representative control C57BL/6 Foxp3-GFP mouse. Quantification is shown as means ± SEM with unpaired *t*-test on the right part of these panels. **c–f** 6–8-week-old SPF or germ-free C57BL/6 mice were injected or not i.p. with 200 µg of anti-LFA-1 (αL) and anti-VLA-4 (α4) Abs. Forty-eight hours later, SLOs were harvested and analyzed. **c** Diagram illustrating the experimental model. **d** Total cell numbers recovered from pLNs, mLNs, and Peyer's patches (Pp) of SPF or germ-free mice. **e** Recovery of CD4 Treg and CD4 Tmem cells in pLNs and mLNs. **f** Recovery of CD4 Treg, CD4 Tmem and CD8 Tmem cells in Peyer's patches (Pp). Each dot represents an individual mouse (unpaired *t*-test). *$p < 0.05$, **$p < 0.01$, ***$p < 0.001$. ns not significant. Mouse clip arts were generated in (21)

Treg cells. Similarly, genes downregulated in CD4 Treg cells from muscle and VAT were mostly less expressed in day 2 LN CD4 Treg cells than in LN CD4 Treg cells from control mice (Fig. 6a and Supplementary Fig. 6b). Similar results were obtained with CD4 Tmem cells (Fig. 6b and Supplementary Fig. 7c).

Out of 327 genes, 109 were differentially expressed between day 2 and control LN CD4 Treg cells, and 53 out of 308 genes for CD4 Tmem cells (for this comparison, data sets were filtered to common probes and a fold change cut-off of ≥1.5 was used) were also part of the transcriptomic profiles of muscle and VAT CD4 Treg and CD4 Tmem cells respectively (Fig. 6c, d). Thus, 33% and 17% of differentially expressed genes in day 2 LN CD4 Treg and Tmem cells respectively, were also part of the transcriptional profile of their muscle and VAT cell-counterparts. Overall, those transcripts distinguished resident CD4 T cells from their

circulating cell-counterparts, and thus represented the core transcriptional signature of resident CD4 Treg and Tmem cells, whatever the tissue (lymphoid or non-lymphoid) they originated from. Among these common genes, several chemokine and cytokine receptor genes were found, including *S1pr1*, *Ccr2*, and *Ccr8* for CD4 Treg cells and *Ccr2*, *Il10ra*, *Il18r1*, and *Il2rb* for CD4 Tmem cells (Fig. 6e, f).

**S1PR1 downregulation accounts for T cell residence in SLOs.** One of the transcripts with a difference in expression between day 2 LN CD4 Treg cells and their cell-counterparts from untreated mice, encodes the receptor S1PR1, which controls the entry of T cells into the lymphatics and thus their exit from SLOs[30]. We therefore decided to analyze its involvement in CD4 T cell

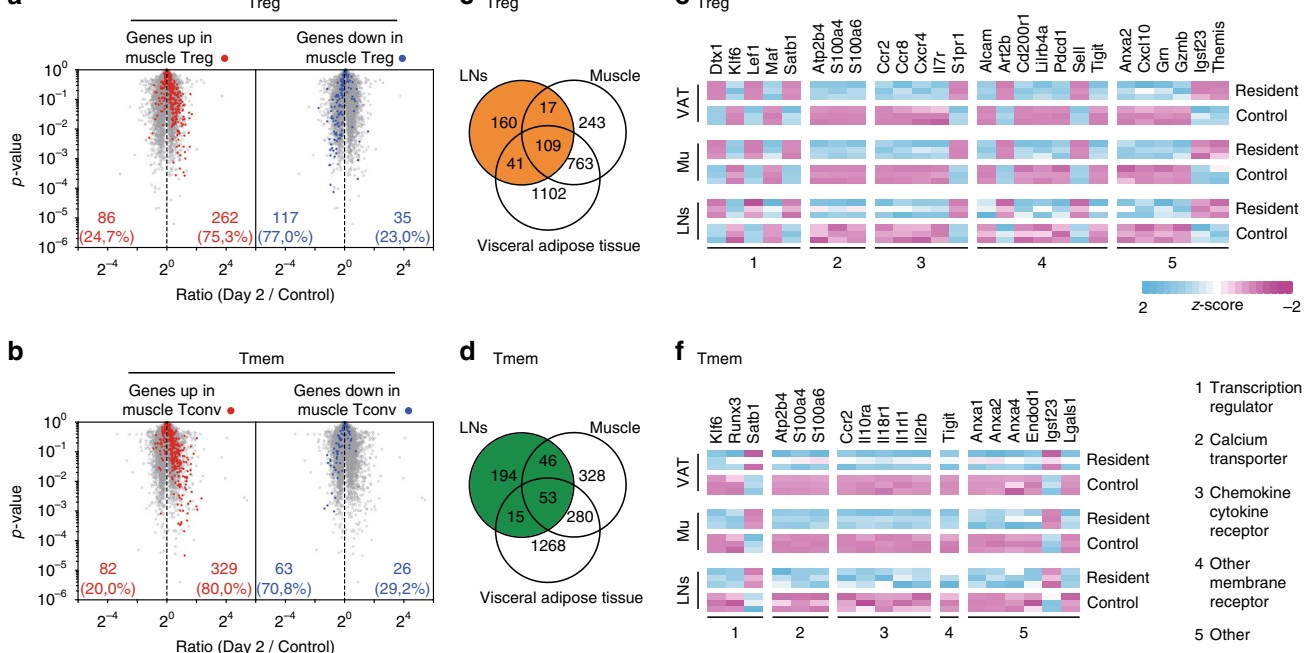

**Fig. 6** Gene expression profiling of LN-resident CD4 Treg and Tmem cells. **a** "Volcano plot" representation (Log₂ (ratio) versus Log₁₀ (t test p value)) between day 2 and control LN CD4 Treg cells from C57BL/6 Foxp3-GFP mice. Among the 500 genes the most differentially expressed between muscle and control CD4 Treg cells with a P value of <0.05, upregulated and downregulated genes are highlighted in red and blue respectively. Data sets were filtered to common probes between the two arrays. **b** Same as in **a** for CD4 Tmem cells. **c** Comparison of the sets of genes differentially expressed between day 2 LN, Muscle and VAT CD4 Tregs cells and their respective control cells. Data sets were filtered to common probes between the three arrays. **d** Same as in **c** for CD4 Tmem cells. **e** Expression pattern of chosen genes differentially expressed (±1.5 fold change, with a p value of <0.05) between day 2 LN, Muscle and VAT CD4 Tregs cells and their respective control cells. **f** Same as in **e** for CD4 Tmem cells

retention in SLOs. It should be noted that S1PR1 expression was also found to be downregulated in day 2 LN CD4 Tmem cells when compared to control CD4 Tmem cells (day2/control = 0.80, p = 0.006 in the transcriptome analysis shown in Fig. 6) although to a lesser extent than observed for CD4 Treg cells (day2/control = 0.49, p = 0.0001).

In our hands, detection of S1PR1 surface expression required a two-step staining that could not be used with cells from VLA-4 and LFA-1 mAb-treated mice, as both S1PR1 and LFA-1 mAbs are rat IgG₂ₐ mAbs. We therefore returned to parabiosis experiments. Based on the expression of S1PR1 and CD69, one can define three subpopulations among CD4 Treg and Tmem cells in SLOs: CD69⁻S1PR1⁺, CD69⁻S1PR1⁻, and CD69⁺S1PR1⁻ cells (Fig. 7a). Enrichment in host cells was clearly observed in CD69⁺ S1PR1⁻ CD4 Treg cells for all SLOs studied, demonstrating a limited exchange of such cells between the two parabionts. The proportion of host cells among CD69⁻S1PR1⁻ cells was also increased, but to a lesser extent, than in CD69⁺S1PR1⁻ cells, whereas full chimerism was almost reached for CD69⁻S1PR1⁺ cells (Fig. 7b). Results were more complex for CD4 Tmem cells as a similar clear enrichment in host cells among CD69⁺S1PR1⁻ CD4 Tmem cells was only observed in the spleen and in Peyer's patches (Fig. 7b).

As described above, the proportion of resident CD4 Treg and Tmem cells within SLOs increased with age (Fig. 3). In accordance with the preferential retention of CD69⁺S1PR1⁻ CD4 Treg cells, we found that CD4 Treg cells from old mice contained far more CD69⁺S1PR1⁻ cells than CD4 Treg cells from young mice (Fig. 7c). A significant increase with age of the proportion of CD69⁺S1PR1⁻ cells among CD4 Tmem cells was also found in the spleen and Peyer's patches (Fig. 7c). Altogether, these results suggest that S1PR1 downregulation is the main mechanism involved in Treg cell residence within SLOs. Other

mechanisms may be involved in the retention of CD4 Tmem cells especially in LNs.

## Discussion
Whereas naive T cells recirculate continuously between SLOs through lymph and blood, it has become clear that a subset of CD8 Tmem cells, named resident memory T cells, do not recirculate through the body, but remain trapped in non-lymphoid tissues in which they provide protection against local secondary infections[1]. These resident memory T cells derived from precursors that entered tissues during the effector phase of immune responses and remained positioned within this compartment. More recently, resident CD4 Tmem cells have been identified in non-lymphoid tissues in which they are also implicated in immune protection against a variety of tissue-tropic pathogens[3]. Finally, resident CD4 Treg cells were found to colonize multiple non-lymphoid tissues, including the gut, skin, lung, VAT, and muscle in which they control inflammatory cells in the vicinity, preventing the organism from incommensurate sterile inflammatory responses[31]. In this paper, our data demonstrate the existence of CD4 Treg and Tmem cells residing in the SLOs of SPF mice. Indeed, parabiosis experiments clearly demonstrated that full chimerism in SLOs between the two parabionts was still not obtained for CD4 Treg and Tmem cells 2 months after surgery (Fig. 1 and Supplementary Fig. 1). Secondly, CD4 Treg and Tmem cells remained trapped in the LNs and Peyer's patches for several days after blocking the entry of blood T cells into SLOs (Fig. 2). Finally, these "SLO-resident T cells" share a core transcriptional signature with their cell-counterparts from non-lymphoid tissues (Fig. 6). Altogether, our results show that T cell long-term residency is not a particular feature of non-lymphoid tissues, as it also applies to SLOs. Interestingly,

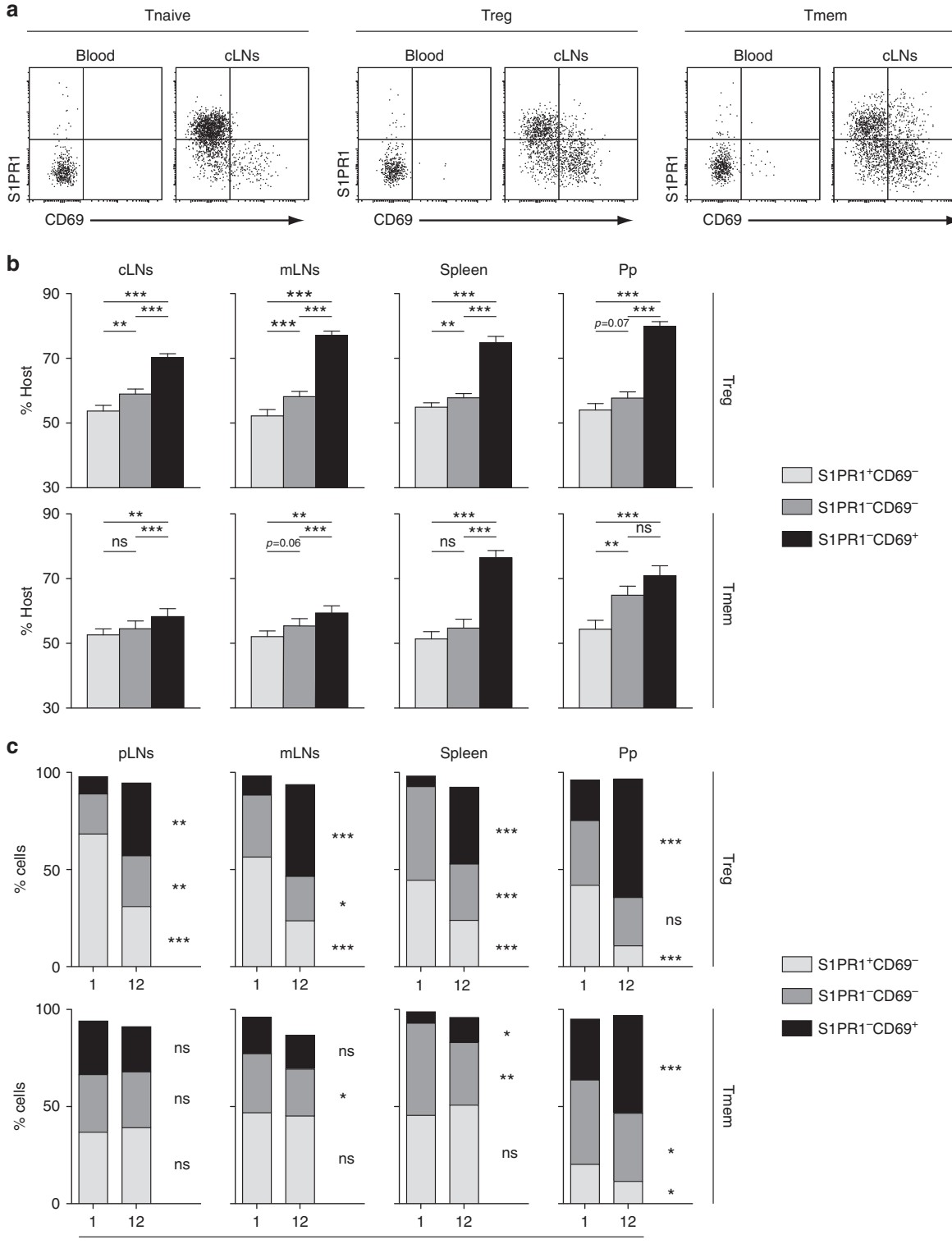

**Fig. 7** S1PR1 downregulation account for T cell residence in SLOs. **a**, **b** Four weeks after parabiosis surgery, blood, cervical LNs (cLNs), mesenteric LNs (mLNs), spleen, and Peyer's patches (Pp) were recovered and analyzed. **a** S1PR1/CD69 representative dot-plots are shown for blood and cLN CD4 Treg, Tmem, and Tnaive cells from a representative parabiont. **b** Proportions of host cells (CD45.1+ for the CD45.1 parabiont and CD45.2+ for the CD45.2 parabiont) among the indicated CD4 Treg (upper panel) and Tmem (lower panel) cell-subsets recovered from cLNs, mLNs, spleen, and Peyer's patches (Pp) are shown as means ± SEM with paired $t$-test for six parabiotic pairs from three independent experiments. **c** pLNs, mLNs, spleen, and Peyer's patches (Pp) of 1- and 12-month-old mice were recovered and analyzed. Proportion of the indicated CD4 Treg (upper panel) and CD4 Tmem (lower panel) cell-subpopulations are shown as means with unpaired $t$-test. *$p < 0.05$, **$p < 0.01$, ***$p < 0.001$. ns not significant

although well documented in numerous non-lymphoid tissues, resident CD8 Tmem cells were absent from the spleen and LNs of SPF mice.

In non-lymphoid tissues, the generation of resident T cells occurs predominantly after primary infections[8]. Indeed, resident memory CD8 T cells in non-lymphoid tissues are enriched in cells specific for a given pathogen, several months after primary infection. Similarly, pathogen-specific T cells have also been described in SLOs[11–13]. We thus first supposed that microbiota-derived antigens may play a crucial role in the generation of resident T cells within the SLOs of SPF mice. Antibiotic treatments and the study of germ-free mice revealed that, with the exception of Peyer's patches, microbiota does not play a major role in establishing and maintaining T cell residency within the SLOs of SPF mice. Thus, in LNs, either T cell residency is not relying on T cell specificity or it involves recognition of self-derived peptides. The phenotype of LN-resident CD4 Treg and Tmem cells with respect to markers whose expression correlates positively (CD5, Nur77) or negatively (Ly-6C) with the ability of a T cell to interact with self antigens, would support the second hypothesis. The existence of self antigen-specific resident CD4 Treg cells within the LNs of SPF and germ-free mice makes sense, since such cells would protect the drained organs from efficient priming of conventional self-reactive T cells, thus preventing autoimmunity. This model corroborates with the results showing the prominent role of CD4 Treg cells from a given draining LN in the prevention of autoimmune diseases, targeting the related organs such as oophoritis, prostatitis and dacryoadenitis[32–34]. Our results also agree with data showing an antigen- and dendritic cell-dependent accumulation of prostate-specific CD4 Treg cells in prostate-draining LNs, an accumulation that can be abrogated upon castration[35,36].

The existence of self antigen-specific resident CD4 Tmem cells in the SLOs of SPF mice is somehow more puzzling. Indeed, in response to infections, naive T cells are primed in SLOs in which they differentiate into effector cells. During this step, they are induced to express chemokine receptors. Then, they migrate to the infected tissues in response to chemokine gradients to fight the pathogens. Finally, once the pathogen is cleared, some of these cells remain in the tissue as tissue resident CD4 Tmem cells and would represent a first line of defense preventing re-infection. In the SLOs of SPF mice, we hypothesize that resident CD4 Tmem cells are generated in response to microbiota and self antigens. In these conditions, the nature of either APC or of the cytokines in their environment may cause differences in their differentiation, particularly in terms of chemokine receptors and that could participate to their trapping in SLOs. In fact, self antigen-specific effector CD4 Tmem cells represent a potential danger for organs. Trapping them in SLOs may prevent them reaching the drained organs, and initiating autoimmunity. CD4 Tmem cell-residence within SLOs may thus correspond to an active tolerance mechanism which prevents the migration of potentially harmful T cells into non-lymphoid tissues. Co-clustering of self antigen-specific CD4 Treg and CD4 Tmem cells has been recently described in LNs by Liu et al.[37] and these cells may correspond to the subsets of SLO resident CD4 Treg and Tmem cells we are describing in this paper.

The membrane receptor S1PR1, by allowing T cells to respond to the sphingolipid S1P present within efferent lymph, is the central mediator of lymphocyte egress from SLOs[30,38,39]. Inhibiting S1P responsiveness, through either downregulation of S1pr1 transcription or CD69-dependent internalization and degradation of membrane S1PR1, appears also to be required for establishing and maintaining T cell residence within non-lymphoid tissues[8,40,41]. Our parabiosis experiments show that among CD4 Treg cells, CD69+S1PR1− cells are strongly enriched

in resident cells in all SLOs (Fig. 7). This is not the case for CD69−S1PR1− cells, perhaps because these cells correspond mostly to cells that have just entered SLOs from the blood. Similar results were found for CD4 Tmem cells in the spleen and Peyer's patches of parabionts, but enrichment was quite moderate in LN CD69+S1PR1− CD4 Tmem cells. Thus, for CD4 Treg cells in all SLOs studied, as well as for CD4 Tmem cells from the spleen and Peyer's patches, disrupting the ability of T cells to respond to S1P gradients may be the main mechanism of T cell retention. Other mechanisms may participate in CD4 Tmem cell retention within LNs. Recently, Zhang et al. and Audemard-Verger et al. suggested that a subset of innate-like αβ and γδT cells, defined as CCR6+CD127hi cells, are trapped within pLNS for long periods of time, and that this retention involves CCR6-dependent migration of these cells to the subcapsular sinus of the LNs and CD169, a sialic acid-binding lectin expressed by subcapsular macrophages[16,42]. Day 2 CD4 Tmem cells from pLNs almost all express low surface levels of CD127 (Fig. 4). Thus, retention of innate-like TCRαβ+ T cells and of conventional CD4 Tmem cells within pLNs certainly relies on different mechanisms.

A constant proportion of Treg cells among CD4 T cells in the blood have been documented in mice and a marginal increase, if any, has been reported in humans whereas the frequency of CD4 Treg cells increases considerably in SLOs (Fig. 3a,[43–46]). Here, we show that the proportion of resident T cells within the CD4 Treg- and Tmem-cell compartments in LNs increases considerably with age (Fig. 3). This observation may help to explain why blood-cell composition in mice poorly reflects the increases with age of the frequencies of CD4 Treg and Tmem cells observed in SLOs. Moreover, an increase with age of resident CD4 Treg cells in SLOs may play an important role in the lower vaccine-induced immune responses and the reactivation of chronic infectious diseases observed in the elderly[43,47].

## Methods

**Mice**. Six to twelve-week-old mice were used for experiments unless otherwise indicated. C57BL/6 Foxp3-GFP CD45.1 or CD45.2 mice[22,48] were maintained in our own animal facilities, under SPF conditions. C57BL/6 germ-free (GF) mice and the related control C57BL/6 SPF mice were obtained from CDTA Orléans (TAAM CNRS UPS44). All procedures were approved by the French animal experimentation and ethics committees of Paris Descartes and Pierre and Marie Curie universities and validated by "Service Protection et Santé Animales, Environnement" with the numbers C-75-562 and A-75-1315. Sample sizes were chosen to assure reproducibility of the experiments in accordance with the replacement, reduction and refinement principles of animal ethics regulation.

**Parabiosis**. Female host parabionts were generated with 8–12-week-old C57BL/6 Foxp3-GFP CD45.1 mice and C57BL/6 Foxp3-GFP CD45.2 mice. The percentage of host cells in a given T cell subset corresponds to the percentage of CD45.1+ cells for the CD45.1 parabiont and the percentage of CD45.2+ cells for the CD45.2 parabiont.

**Blocking T cell entry into LNs and Pp**. C57BL/6 Foxp3-GFP mice were injected or not i.p. with 200 µg of anti-LFA-1 (clone M17/4) and anti-VLA-4 (clone PS/2) Abs (BioXcell) to block T cell entry into LNs and Peyer's patches as previously described[22].

**Antibiotic treatment**. For ablation of commensal microbiota, an antibiotic cocktail of 1 g/l each of Ampicillin (sodium salt, Teva), Neomycin sulfate (Sigma), Metronidazole (Sanofi), and 0.5 g/l Vancomycin hydrochloride (Fisher) was used[49]. Antibiotics were added to the drinking water on a weekly basis for 4 weeks.

**Cell surface staining and flow cytometry**. SLOs were homogenized and passed through a nylon cell strainer (BD Falcon) in 5% FCS, 0.1% NaN3 (Sigma-Aldrich) in phosphate saline buffer saline (PBS). Cell suspensions were then dispensed into 96-well round-bottom microtiter plates. Surface staining was performed as previously described[50,51]. For determination of intracellular cytokine production, cells were stimulated with 0.5 µg/ml PMA, 0.5 µg/ml ionomycin, and 10 µg/ml brefeldin A (all from Sigma-Aldrich) for 2 h at 37 °C. Cells were then stained for surface markers, fixed in 2% paraformaldehyde in PBS, and permeabilized with 0.5% saponin, followed by labeling with specific cytokine Abs. This fixation and

permeabilization protocol was also used to determine intra+extracellular expression of CCR7. The Foxp3 Staining Buffer Set (eBioscience) was used for Foxp3, Nur77, CTLA-4, and Ki-67 intracellular staining. S1PR1 was detected with a rat monoclonal antibody (clone 713412, R&D Systems). Briefly, cells were first incubated for 90 min with the anti-S1PR1 rat mAb (50 μg/ml) in PBS containing 0.5% FCS, 1 mM EDTA, 0.05% azide, and 2% normal mouse serum. After washing, the cells were then incubated with donkey biotinylated anti-rat IgG polyclonal Abs (Jackson ImmunoReaserch) for 30 min in the above medium followed by streptavidin-APC (BioLegend). Multi-color immunofluorescence was analyzed using a BD-LSR2 or a BD-Fortessa cytometer (BD Biosciences). Data acquisition and cell sorting were performed at the Cochin Immunobiology facility.

**Microarray.** CD4 T cells from LNs (pooled peripheral and mesenteric LNs) of C57BL/6 Foxp3-GFP mice were enriched as previously described[22]. Then, CD4 Treg and Tmem cells were flow-cytometry sorted as defined in the gating strategy shown in Fig. 1b using a FACS-ARIA3 flow cytometer. Total RNA was extracted using the RNeasy Mini kit (Cat. No./ID: 74104, QIAGEN). RNA quality was validated with Bioanalyzer 2100 (using Agilent RNA6000 nano chip kit). Experimental and analytical parts of the microarray analysis were performed according to the MIAME standards. Amplified, fragmented and biotinylated sense-strand DNA targets were synthesized from 100 ng total RNA according to the manufacturer's protocol (Genechip Whole transcript (WT) Sense Target labeling assay kit (Affymetrix)) and hybridized to a mouse gene 1.0 ST array (Affymetrix). The stained chips were read and analyzed with a GeneChip Scanner 3000 7G and Expression Console software (Affymetrix). Raw data (.cel files) were then processed and normalized using the quantile normalization method in RMA with R package (Bioconductor). Statistical analysis was then performed with MEV software (TIGR). Microarrays were performed at the Cochin Genom'ic facility.

**Calculations.** For Fig. 3c, d, calculations were done as follows: for one given experiment, % circulating Treg cells among CD4 T cells = 100×(absolute number of CD4 Treg cells in control mice−absolute number of day 2 CD4 Treg cells)/ (absolute number of total CD4 T cells in control mice−absolute number of day 2 CD4 Treg and Tmem cells); % circulating Tmem cells among CD4 T cells = 100×(absolute number of CD4 Tmem cells in control mice−absolute number of day 2 CD4 Tmem cells)/(absolute number of total CD4 T cells in control mice −absolute number of day 2 CD4 Treg and Tmem cells); % circulating naive cells among CD4 T cells = 100−% circulating Treg and Tmem cells.

**Statistics.** Data are expressed as mean ± SEM, and the significance of differences between two series of results was assessed using the Student's unpaired or paired t test. Values of $p < 0.05$ were considered as statistically significant (*$p < 0.05$; **$p < 0.01$; ***$p < 0.001$).

**Data availability.** The original data discussed in this publication have been deposited in the Gene Expression Omnibus at http://www.ncbi.nlm.nih.gov/geo/ (accession number GSE104011). The authors declare that the data supporting the findings of this study are available within the article and its supplementary information files, or are available upon reasonable requests to the authors.

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

## Acknowledgements

We greatly acknowledge E. Maillard, K. Labroquère and M. Andrieu from the Cochin Immunobiology facility, S. Jacques and F. Letourneur from the Cochin Genomic facility. This work was supported by grants from the "Fondation pour la Recherche Médicale" (FRM team), the "Ligue contre le Cancer" (from the Ile de France committee), the "Association pour la Recherche sur le Cancer" and the "French National Research Agency" (grant ANR-15-CE15-0009-01). A. Delpoux was supported by a Ph.D. fellowship from the "Association pour la Recherche sur le Cancer". A. Audemard and P. Hamon were supported by a Ph.D. fellowship from the "Ligue contre le Cancer" and from INSERM.

## Author contributions

A.D., A.A.-V., and B.L. designed experiments. A.D., A.A.-V., V.G., R.M., A.D., P.H., N.B., M.R., J.D., B.M., C.A., and A.B. did the experiments. A.D. and A.A.-V. analyzed the flow cytometry data. V.G. and C.A. analyzed the transcriptomic data. A.D., V.G., and C.A. made the figures. B.L. wrote the paper.

## Additional information

**Competing interests:** The authors declare no competing financial interests.

