## [Peer Review File · Nature Communications]

Reviewers' comments:

Reviewer #1 (lymphocyte trafficking)(Remarks to the Author):

Using a parabiosis approach and a LN entry blockage approach, this study measures the fraction of Treg and Tmem cells in LNs that are resident. The existence of resident T cells in LNs has been demonstrated in a number of past studies but this work represents an advance in carefully quantitating the amount of resident cells. It shows (perhaps not surprisingly) that the number of resident cells increases with age. It shows (consistent with other findings about resident cells) that the LN resident cells have pre-committed effector functions and the cells have elevated Nur77 and CD5 and lower levels of Ly6C. Interestingly, the fraction of resident cells in PPs but not LNs is influenced by the microbiome. Gene expression analysis shows that the LN resident cells are similar to resident Treg and Tmem cells in fat and muscle. Finally, LN residence is shown to correlate with low S1PR1 expression and high CD69 expression, in accord with the established role of S1PR1 in promoting LN egress. Overall, while this study does not contain major surprises, it is valuable in revealing the amount of resident CD4 and CD8 T cells in LNs and PPs and some of the requirements for this residence. The study does not establish the importance of these resident cells.

One specific concern is whether the a4 and aL antibody engagement of the T cells contributes to any of the phenotypes (in particular those only shown by this technique and not confirmed by parabiosis). For example, can we be sure that cell engagement with integrin binding antibodies does not directly augment IL10 or IL17 expression? This could be tested by coating cells in the control condition with the antibodies just prior to the assay.

Minor: the % staining for IL5 (Fig. 4c) is so small the data are not convincing, making the IL5 data in Fig. 4d potentially misleading.

It should be indicated how much the total cell recovery from PP (and LN) is affected by the antibiotic treatment or from SPF vs GF mice.

The authors should ensure that all studies that have reported examples of resident T cells in LNs are cited. e.g. Marriott et al., EJI 47, 660 (2017); Zhang et al., eLife 2016 e18156.

Reviewer #2 (Treg, TCR signalling)(Remarks to the Author):

This is a clear, well written paper that demonstrates that memory CD4⁺ T cells and Treg are retained in many secondary lymph node organs while memory CD8⁺ cells are only retained in Peyer's patches. The authors use two very different approaches that complement each other nicely to illustrate their findings. The first approach is analysis of cell trafficking using parabiosis and the second is an analysis of cells remaining in secondary lymphoid organs when the entry of naïve cells is blocked.

Comments

1. Although the authors clearly demonstrate that the results of their two approaches yield very similar data, I find the rationale between the studies in figure 4 somewhat confusing. Presumably, the pool of memory cells in the SLO changes very little other than being enriched in percentage versus naïve cells. Why should one observe enrichment in the percentage of effector type T cells assuming one is gating on the same CD44^{hi} population of cells before and after the antibody blockade. Is there a difference in CD44 expression patterns in the two groups? Do some cells with a less activated phenotype that are present in the SLO as memory cells before antibody treatment actually exit the SLO during the two days of blockade. To put the question differently—what determines when a memory T cell actually exits and recirculates and when it becomes a permanent resident. The authors need to address this issue. Clearly, a recently primed CD4⁺ T cells that is CD44^{hi} and is now an effector cell must leave the priming environment to exert its effector cell function.
2. The authors do not adequately address the issue of memory versus naïve Treg cells. The Campbell lab has defined these based largely on CD44 expression, while the Lucas group has used the expression of Ly-6C as a differential marker. Some of this data is shown in Figure 5c and 5d in their attempt to relate it to requirements of the Ly-6C⁻ subset for MHC class II triggering which has only been shown in MHC class II⁻ mice). This data belongs in Figure 1 as it appears that both memory Treg and memory T conventional cells are retained in SLOs.
3. It is very difficult to determine absolute numbers from the data presentation in Figure 2f (presented as percent recovery).
4. The data on S1PR1 expression is entirely correlative in nature and the conclusions drawn from this data toned down in the absence of studies demonstrating involvement of this pathway using inhibitors.
5. On page 10, the authors should state that they are measuring receptor expression by PCR (?).
6. One thing missing from this paper is a study of an antigen-specific CD4⁺ memory T cell specific for a foreign antigen. Would such a T cell be retained in SLOs. Although it is widely accepted that CD44^{hi} T cells in the unprimed mouse are antigen-specific memory cells, very little hard data is available to support this view. For example, this could be tested after LCMV-infection where a tetramer is available that readily detects expanded antigen-specific CD4⁺ T cells. Indeed, the Paul lab (Plos Biology e1001171, 2011) has shown that antigen-specific memory CD4⁺CD44^{hi} cells proliferate in vivo at much slower rates than the non-antigen-specific

CD4+CD44hi population.

Reviewer #3 (TCR repertoire, tissue resident T)(Remarks to the Author):

This paper reports the existence of “resident” CD4 T cells of the Treg and Tmem types which hold long-term residence in secondary lymphoid organs, thus not solely in non-lymphoid tissues as previously thought. The work is well thought out, competently and rigorously done, the conclusion is novel and well-discussed, and this reviewer supports publication in Nature Communications, with a few edits.

Minor points:

- Much of the data rests on the effect of treatment with anti-a4 and anti-aL, complemented by parabiotic experiments, and the treatment effect is solely interpreted as affecting cell migration. One would like to see a caveat that this antibody blockade might have consequences other than on cell mobility (e.g. phenotypic changes in the cells, transcriptional induction) that might affect the experimental outcome.
- The antibiotic treatment experiments are questionable (such treatments do not really remove bacteria but rather induce a massive change in flora composition; and bacterial antigens may well persist. The germfree experiments should be used, rather than relegated to Supplement
- The notion that the cells “reside permanently” (p6 and elsewhere) is overstated and should be edited out. The data show slow mobility, not permanence.
- Although not a condition for acceptance, the authors might make their story more interesting if they had a means to localize the resident cells in the SLOs (in line with the Liu paper?). Do they reside in usual paracortex T cell areas, or elsewhere?
- p3, end first paragraph: resident Treg cells have broader effects than just dampening T-cell responses, they talk to other immune and non-immune cell-types.
- Could quote the recent work on skin Tregs (Cell 2017)
- The English is good; on p3, “outnumber the number of their circulating counterparts”, delete the redundant “the number of”. A few gallicisms here and there.

Responses to Reviewers:

Reviewer #1:

Reviewer 1 found that our data represent an advance over previous publications in carefully quantifying the amount of resident cells in the secondary lymphoid organs of SPF mice in the steady state. We reported valuable data revealing the amount of resident CD4 and CD8 T cells. However, at that time, reviewer 1 pointed out several concerns to improve our study. We are pleased to have the opportunity to address the concerns of this referee.

“One specific concern is whether the a4 and aL antibody engagement of the T cells contributes to any of the phenotypes (in particular those only shown by this technique and not confirmed by parabiosis). For example, can we be sure that cell engagement with integrin binding antibodies does not directly augment IL10 or IL17 expression? This could be tested by coating cells in the control condition with the antibodies just prior to the assay.”

We agree with the reviewer that we cannot completely exclude that neutralizing LFA-1 and VLA-4 integrins may have some impact on T-cell biology. It is the reason why we have tried all along this study to confirm the key results obtained using this technique by results obtained through the analysis of parabiotic mice. Furthermore, as clearly shown on the initial supplementary figure 4 (Supplementary figure 5 in the revised version of our manuscript), the increase in the expression of key molecules involved in the suppressive capacities of Treg cells by day 2 LN CD4 Treg cells was counterbalanced by a decrease in the expression of these same molecules by spleen CD4 Treg cells. These results suggest strongly that in response to integrin neutralization, most CD4 Treg cells with low expression of these molecules have exited LNs and accumulated in the spleen. Indeed, if these changes would have resulted directly from the engagement of LFA-1 and VLA-4 by the corresponding blocking antibodies, one should expect to observe similar phenotypic variations in all SLOs, including the spleen.

As requested by the reviewer, we are now showing that integrin engagement by monoclonal antibodies does not potentiate the ability of T cells to produce cytokines (see our new supplementary figure 4). The text of the revised version of our manuscript has been modified accordingly:

“Of note, integrin engagement by monoclonal antibodies prior to re-stimulation with PMA and ionomycin does not potentiate the ex vivo ability of T cells to produce cytokines (Supplementary Fig. 4).”

Minor points

1- “the % staining for IL5 (Fig. 4c) is so small the data are not convincing, making the IL5 data in Fig. 4d potentially misleading.

We agree with the reviewer that very few IL-5-producing CD4 T cells were detected within the SLOs of SPF mice in the steady state. Due to this paucity, the related results may be considered as not convincing and potentially misleading. In the present version of our manuscript, we have thus decided to remove the data on IL-5 production by LN-resident memory CD4 T cells.

2- “It should be indicated how much the total cell recovery from PP (and LN) is affected by the antibiotic treatment or from SPF vs GF mice.”

We are now showing the total cell numbers in the pLNs, mLNs and Peyer’s patches of germ free or SPF mice in figure 5d and those from antibiotic treated or untreated mice in supplementary figure 6d. The text of the revised version of our manuscript has been modified accordingly:

“To determine the relative contribution of microbiota- and self antigen-derived peptides in the retention of T cells within SLOs, we first studied germ free mice (Fig. 5c). The total cell numbers in pLNs and mLNs were comparable between Germ free and SPF mice whereas the cellularity of Peyer’s patches was strongly reduced in germ free mice when compared to SPF mice (Fig. 5d). In agreement, the recovery of day 2 CD4 Treg and Tmem cells was only slightly reduced in pLNs and not affected in mLNs (Fig. 5e). By contrast, 2 days after mAb injection, cell recovery was strongly diminished in the Peyer’s patches of germ free mice when compared to control mice for both CD4 and CD8 Tmem cells (Fig. 5f). Although not significant, a similar tendency seems to apply to CD4 Treg cells. These results were largely confirmed in mice treated with a combination of antibiotics for 4 weeks before blocking T-cell entry into LNs and Peyer’s patches (Supplementary Fig. 6c-f). In this setting, although the total cellularity as well as the absolute numbers of CD4 Treg and Tmem cells were slightly reduced in the pLNs of antibiotic treated mice, those of day 2 CD4 Treg and Tmem cells were not affected by the antibiotic treatment in both pLNs and mLNs. 2 days after mAb injection, cell recovery was strongly and significantly diminished in Peyer’s patches for CD4 Treg cells and for both CD4 and CD8 Tmem cells (Supplementary Fig. 6f). Microbiota thus seems to have an important impact on T-cell residence in Peyer’s patches but only a small or no role in LNs.”

3- “The authors should ensure that all studies that have reported examples of resident T cells in LNs are cited. e.g. Marriott et al., EJI 47, 660 (2017); Zhang et al., eLife 2016 e18156.”

The article by Zhang et al. was already quoted in the introduction and the discussion of our manuscript. We have now cited the recent paper of Marriott et al. in the introduction:

“Photoconvertible fluorescence approaches in mice have suggested the existence of a subset of effector CD4 Tmem cells retained in SLOs that accumulated after immunization or in response to chronic antigen exposure¹⁴,¹⁵,¹⁶,¹⁷. These included follicular helper CD4 T cells¹⁵,¹⁷ and a population of innate-like T cells expressing CCR6 and high surface levels of CD127¹⁶.”

Reviewer #2:

Reviewer 2 found our manuscript clear and well written. However, at that time, reviewer 2 was thinking that our manuscript had significant shortcomings with regard to the presentation of the data and the conclusions drawn from them. We thank the reviewer for this push, because we feel it has allowed us to produce a much stronger manuscript.

1. “Although the authors clearly demonstrate that the results of their two approaches yield very similar data, I find the rationale between the studies in figure 4 somewhat confusing. Presumably, the pool of memory cells in the SLO changes very little other than being enriched in percentage versus naïve cells. Why should one observe enrichment in the percentage of effector type T cells assuming one is gating on the same CD44hi population of cells before and after the antibody blockade. Is there a difference in CD44 expression patterns in the two groups? Do some cells with a less activated phenotype that are present in the SLO as memory cells before antibody treatment actually exit the SLO during the two days of blockade. To put the question differently—what determines when a memory T cell actually exits and recirculates and when it becomes a permanent resident. The authors need to address this issue. Clearly, a recently primed CD4+ T cells that is CD44hi and is now an effector cell must leave the priming environment to exert its effector cell function.”

We agree with the reviewer that during an infection, naive T cells are primed in SLOs in which they differentiate into effector cells which type depends on the cytokine milieu in which they are immersed at the time of their activation. During this step, they are induced to express chemokine receptors that will address them to tissues. Then, they migrate to the infected tissues in response to the corresponding chemokine gradients to fight the pathogens. In our case, we are studying the pool of memory CD4 T cells from Specific Pathogen Free mice in the steady state. We hypothesize that these cells are generated in response to microbiota and Self-derived antigens. It could be that in these conditions, the nature of either antigen presenting cells or of the cytokines in their environment would lead to differences in their differentiation, particularly in terms of expression of chemokine receptors that would participate to their trapping in SLOs. Accordingly, the transcriptomic signatures of resident memory T cells from lymphoid and non-lymphoid tissues reveal that VAT- and muscle-resident CD4 Tmem cells exhibit a significant variation in the transcription of many more chemokine receptors than their cell-counterparts from LNs (See the table below). To address this concern of the reviewer and make our manuscript clearer, we have added a paragraph in the discussion:

“The existence of Self-specific resident CD4 Tmem cells in the SLOs of SPF mice is somehow more puzzling. Indeed, in response to infections, naive T cells are primed in SLOs in which they differentiate into effector cells. During this step, they are induced to express chemokine receptors. Then, they migrate to the infected tissues in

Inserm U1016 • CNRS UMR8104 • Université Paris Descartes
U1016@inserm.fr • www.institutcochin.fr • @InstitutCochin

response to chemokine gradients to fight the pathogens. Finally, once the pathogen is cleared, some of these cells remain in the tissue as tissue resident CD4 Tmem cells and would represent a first line of defense preventing re-infection. In the SLOs of SPF mice, we hypothesize that resident CD4 Tmem are generated in response to microbiota and Self-derived antigens. In these conditions, the nature of either antigen presenting cells or of the cytokines in their environment may lead to differences in their differentiation, particularly in terms of chemokine receptors and that could participate to their trapping in SLOs.”

Symbol	GENEID	LNs		VAT		Muscle	
		p	F R/C	p	F R/C	p	F R/C
Ccr1	12768	0,0930	1,12	0,5032	1,08	0,3096	1,18
Ccr2	12772	0,0169	3,03	0,0191	9,42	0,0004	8,90
Ccr3	12771	0,7033	0,94	0,3224	1,15	0,1771	1,34
Ccr4	12773	0,0603	1,49	0,0635	3,62	0,0047	2,49
Ccr5	12774	0,3493	0,60	0,0460	6,96	0,0000	4,51
Ccr6	12458	0,0418	1,76	0,2115	1,12	0,0039	1,94
Ccr7	12775	0,4435	0,89	0,1066	0,70	0,0049	0,68
Ccr8	12776	0,7723	1,13	0,1462	2,27	0,0050	5,96
Ccr9	12769	0,0888	0,58	0,5809	0,93	0,0125	0,58
Ccr10	12777	0,0275	3,02	0,0180	0,77	0,2296	1,14
Cxcr1	227288	0,3077	0,89	0,0494	0,69	0,3651	0,88
Cxcr2	12765	0,2906	0,81	0,0044	1,12	0,9157	0,98
Cxcr3	12766	0,4850	1,11	0,0094	11,92	0,0002	3,78
Cxcr4	12767	0,4611	1,18	0,0025	3,17	0,0004	1,69
Cxcr5	12145	0,4604	1,25	0,0274	3,95	0,0003	0,33
Cxcr6	80901	0,0502	1,40	0,0219	7,39	0,0007	6,78
Xcr1	23832	0,9501	1,01	0,5895	1,04	0,8323	0,97
Cx3cr1	13051	0,1417	1,39	0,2140	1,29	0,0110	1,64
Cmklr1	14747	0,8771	0,98	0,9216	1,02	0,1606	1,22
Ccr1l1	12770	0,7042	1,02	0,3197	0,89	0,3458	0,87
Ccrl2	54199	0,7599	1,11	0,1311	1,60	0,0002	2,96

F R/C: Fold change between the indicated Resident and Control memory CD4 T cells

2. “The authors do not adequately address the issue of memory versus naïve Treg cells. The Campbell lab has defined these based largely on CD44 expression, while the Lucas group has used the expression of Ly-6C as a differential marker. Some of this data is shown in Figure 5c and 5d in

their attempt to relate it to requirements of the Ly-6C⁻ subset for MHC class II triggering which has only been shown in MHC class II⁻ mice). This data belongs in Figure 1 as it appears that both memory Treg and memory T conventional cells are retained in SLOs.”

We thank the reviewer for this push as it has allowed a better and clearer presentation of our data. As suggested by the reviewer, the results showing that, in parabiosis experiments, CD4 Treg cells retained within SLOs are Ly-6C⁻ (i.e memory Treg cells), are now shown in figure 1f. A similar analysis has also now been performed by using the expression level of CD44 to discriminate between naïve (CD44^{low}) and memory (CD44^{high}) CD4 Treg cells (Supplementary Fig. 1b). The text of the revised version of our manuscript has been modified accordingly:

“We have recently shown that Ly-6C expression allows to discriminate between memory and naïve CD4 Treg cells with Ly-6C⁻ cells corresponding to memory CD4 Treg cells²². Using this marker, we then tested whether the unequal redistribution of CD4 Treg cells between the 2 parabionts was differential between the naïve and memory subpopulations. Interestingly, we found that among CD4 Treg cells, only Ly-6C⁻ CD4 Treg cells were significantly enriched in cells deriving from their host in all SLOs tested (Fig. 1f). Similar results were obtained by using, as proposed by Smigiel et al.²³, the expression level of CD44 to discriminate between naïve (CD44^{low}) and memory (CD44^{high}) CD4 Treg cells (Supplementary Fig. 1b). Altogether, these results suggest that a significant proportion of memory regulatory and conventional T cells are residing for at least several weeks within SLOs in the steady state.”

The expression of Ly-6C by day 2 Treg cells has been incorporated in supplementary figure 5 and the text of our manuscript modified accordingly:

“Finally, we studied more extensively the phenotype of day 2 CD4 Treg cells (Supplementary Fig. 5). We first confirmed the memory phenotype of these cells as they were expressing high surface amounts of CD44 and were not expressing Ly-6C⁺^{22,23}. We then examined the expression of molecules known to play a role in the suppressive capacities of CD4 Treg cells.”

3. “It is very difficult to determine absolute numbers from the data presentation in Figure 2f (presented as percent recovery).”

In the present version of our manuscript, we are now showing, in supplementary figure 2c, the absolute numbers of Tregs cells and of naïve and memory conventional T cells as a function of time after LFA-1 and VLA-4 neutralization.

4. “The data on SIPRI expression is entirely correlative in nature and the conclusions drawn from this data toned down in the absence of studies demonstrating involvement of this pathway using inhibitors.”

We agree with the reviewer that our data on the role S1PR1 in the retention of CD4 T mem and CD4 Treg cells within SLOs are mostly correlative. However, it is important to note that our conclusions are supported nonetheless by the fact that S1PR1⁻ cells are enriched in non-circulating cells in parabionts but also by our data showing that the proportion of S1PR1⁻ cells increases with age. Pharmaceutical inhibitors of the S1PR1/S1P axis have been described for a long time. FTY-720 persistently activates and chronically down-regulates S1PR1 and 4-deoxyripyridoxine (DOP) annihilates the gradient required for lymphocyte egress from SLOs into efferent lymphatic by stabilizing S1P through the inhibition of the S1P lyase activity. We have studied the effect of this latter drug on the recovery of naive, regulatory and memory cells 2 days after anti- α 4 and anti- α L mAb injection and, as expected, T-cell retention within SLOs was greatly increased with even naïve T cells being trapped within SLOs.

5. “On page 10, the authors should state that they are measuring receptor expression by PCR (?)”

Numbers indicated on page 11 correspond actually to the average of the values obtained in the transcriptome analysis presented in figure 6. This point has been clarified in the revised version of our manuscript.

6. “One thing missing from this paper is a study of an antigen-specific CD4+ memory T cell specific for a foreign antigen. Would such a T cell be retained in SLOs. Although it is widely accepted that CD44hi T cells in the unprimed mouse are antigen-specific memory cells, very little hard data is available to support this view. For example, this could be tested after LCMV-infection where a tetramer is available that readily detects expanded antigen-specific CD4+ T cells. Indeed, the Paul lab (Plos Biology e1001171, 2011) has shown that antigen-specific memory CD4+CD44hi cells proliferate in vivo at much slower rates than the non-antigen-specific CD4+CD44hi population.”

We agree with the reviewer that to define the specificity of SLO-resident CD4 Treg and CD4 Tmem cells in SPF mice would be very interesting. In the current manuscript, antibiotic treatments and the study of germ-free mice revealed that, with the exception of Peyer’s patches, microbiota does not play a major role in establishing and maintaining T-cell residency within the SLOs of SPF mice. Thus, in these conditions, T-cell residency certainly involves recognition of Self-derived peptides in LNs.

To study whether infections may lead to the generation of resident T cells within SLOs is of course of interest. Some articles already provide clues concerning that point. Indeed, Takamura et al. have shown the retention of CD8 Tmem cells within draining mediastinal LNs after lung infections (JEM 2010). Using TCR-transgenic cells (P14 and OTI), Schenkel et al. have documented the existence of antigen-specific CD8 Tmem cells within the spleen and LNs after LCMV and VSV-OVA acute infection in mice (JI 2014). Finally, using MHCII tetramers, Marriot et al. have recently highlighted the presence of antigen-specific CD4 Tmem cells residing in the draining LN for months following immunization (EJI 2017). All of these references have been quoted in the introduction of our manuscript. It would now be very interesting to study T-cell residency in “dirty mice” (Masopust et al. JI 2017) and to compare the results with our results obtained in SPF mice.

In our study, mice were not infected, nor immunized. However, in these conditions, SLO resident CD4 Treg cells and CD4 Tmem cells (but not CD8 Tmem cells except in Peyer’s patches) can be detected and we have here quantified and characterized these cells. We are now of course planning to determine whether these resident T cells (or at least a part of them) are “specific” of the SLO in which they are trapped. In other words, are the CD4 Treg and Tmem cells trapped in the mesenteric LNs the same than the ones residing in the inguinal LNs? But it will be a long story involving, among others, repertoire studies.

Reviewer #3:

Reviewer 3 found that we reported intriguing and well-presented data and that we were providing novel insights on the behavior of $\alpha\beta$ T cells in secondary lymphoid organs. However, at that time, reviewer 3 pointed out several minor points to improve our study. We thank the reviewer for this push, because we feel it has allowed us to produce a much stronger manuscript.

1- “Much of the data rests on the effect of treatment with anti-a4 and anti-aL, complemented by parabiotic experiments, and the treatment effect is solely interpreted as affecting cell migration. One would like to see a caveat that this antibody blockade might have consequences other than on cell mobility (e.g. phenotypic changes in the cells, transcriptional induction) that might affect the experimental outcome.”

We agree with the reviewer that neutralizing integrins may have consequences on T-cell biology other than blocking T-cell entry into lymph-nodes and Peyer’s patches. It is the reason why we have tried all along this study to confirm the key results obtained using this technique by results obtained through the analysis of parabiotic mice. Furthermore, as clearly shown on the initial supplementary figure 4 (Supplementary figure 5 in the revised version of our manuscript), the increase in the expression of key molecules involved in the suppressive capacities of Treg cells by day 2 LN CD4 Treg cells was counterbalanced by a decrease in the expression of these same molecules by spleen CD4 Treg cells. These results suggest strongly that in response to integrin neutralization, most CD4 Treg cells with low expression of these molecules have exited LNs and accumulated in the spleen. Indeed, if these changes would have resulted directly from the engagement of LFA-1 and VLA-4 by the corresponding blocking antibodies, one should expect to observe similar phenotypic variations in all SLOs, including the spleen. Similar results were observed for the markers shown in the other figures, such as CD62L, CD127 and CCR7 in Figure 4.

2- “The antibiotic treatment experiments are questionable (such treatments do not really remove bacteria but rather induce a massive change in flora composition; and bacterial antigens may well persist. The germfree experiments should be used, rather than relegated to Supplement

In the revised version of our manuscript, we have now interchanged the “germfree” and the “antibiotic treatment” experiments. The “germfree” experiments are now shown in figure 5 and the “antibiotic treatment” experiments in supplementary figure 6. The text of our

manuscript has been modified accordingly:

“To determine the relative contribution of microbiota- and self antigen-derived peptides in the retention of T cells within SLOs, we first studied germ free mice (Fig. 5c). The total cell numbers in pLNs and mLNs were comparable between Germ free and SPF mice whereas the cellularity of Peyer’s patches was strongly reduced in germ free mice when compared to SPF mice (Fig. 5d). In agreement, the recovery of day 2 CD4 Treg and Tmem cells was only slightly reduced in pLNs and not affected in mLNs (Fig. 5e). By contrast, 2 days after mAb injection, cell recovery was strongly diminished in the Peyer’s patches of germ free mice when compared to control mice for both CD4 and CD8 Tmem cells (Fig. 5f). Although not significant, a similar tendency seems to apply to CD4 Treg cells. These results were largely confirmed in mice treated with a combination of antibiotics for 4 weeks before blocking T-cell entry into LNs and Peyer’s patches (Supplementary Fig. 6c-f). In this setting, although the total cellularity as well as the absolute numbers of CD4 Treg and Tmem cells were slightly reduced in the pLNs of antibiotic treated mice, those of day 2 CD4 Treg and Tmem cells were not affected by the antibiotic treatment in both pLNs and mLNs. 2 days after mAb injection, cell recovery was strongly and significantly diminished in Peyer’s patches for CD4 Treg cells and for both CD4 and CD8 Tmem cells (Supplementary Fig. 6f). Microbiota thus seems to have an important impact on T-cell residence in Peyer’s patches but only a small or no role in LNs.”

3- “The notion that the cells “reside permanently” (p6 and elsewhere) is overstated and should be edited out. The data show slow mobility, not permanence.”

We agree with the reviewer that to write “reside permanently” is somehow overstated. We have now replaced “reside permanently” by “reside for long periods of time” throughout the manuscript.

4- “Although not a condition for acceptance, the authors might make their story more interesting if they had a means to localize the resident cells in the SLOs (in line with the Liu paper?). Do they reside in usual paracortex T cell areas, or elsewhere?”

We agree with the reviewer that it would be very interesting to localize resident regulatory and memory CD4 T cells within SLOs. Unfortunately, to date, we have not identified a reliable marker that would allow us to visualize them within the SLOs of untreated mice. We have also tried to purify LN resident cells 2 day after integrin neutralization, label them with fluorescent dyes and transfer them into untreated recipients. Unfortunately, they did not home back efficiently to LNs as most of them are not expressing CD62L. Moreover, we do not think that to visualize regulatory T cells within LNs 2 days after treatment to be informative as at this time point, more than 95% of B and T cells have exited LNs and it would be very hard to interpret the results.

5- *“p3, end first paragraph: resident Treg cells have broader effects than just dampening T-cell responses, they talk to other immune and non-immune cell-types.”*

In the revised version of our manuscript, we have taken into account this concern of the reviewer and precised that resident regulatory T cells participate to non lymphoid tissue homeostasis through crosstalk with immune and non-immune cells. The introduction of our manuscript has been modified accordingly:

“whereas resident CD4 Treg cells may ensure tissue integrity by dampening T-cell responses to self and commensal bacteria derived antigens and controlling tissue homeostasis through crosstalk with immune and non-immune cells ^{6,7,8}.”

6- *“Could quote the recent work on skin Tregs (Cell 2017)”*

In our manuscript, we are now quoting this reference in the introduction:

“In particular, it has been recently shown that skin resident CD4 Treg cells are required for skin wound healing and hair regeneration through their ability to crosstalk with hair follicle stem cells ^{9,10}.”

7- *“The English is good; on p3, “outnumber the number of their circulating counterparts”, delete the redundant “the number of”. A few gallicisms here and there.”*

This gallicism has been corrected. We have carefully reviewed the manuscript to find others and correct them.

Reviewers' Comments:

Reviewer #1 (Remarks to the Author):

The authors have paid close attention to the reviewer comments in making their revisions. My concerns have been adequately addressed.

Reviewer #2 (Remarks to the Author):

none